# Interaction Insight of Pullulan-Mediated Gamma-Irradiated Silver Nanoparticle Synthesis and Its Antibacterial Activity

**DOI:** 10.3390/polym13203578

**Published:** 2021-10-17

**Authors:** Mohd Shahrul Nizam Salleh, Roshafima Rasit Ali, Kamyar Shameli, Mohd Yusof Hamzah, Rafiziana Md Kasmani, Mohamed Mahmoud Nasef

**Affiliations:** 1School of Chemical Engineering, College of Engineering, Universiti Teknologi MARA Cawangan Terengganu, Bukit Besi Campus, Dungun 23200, Terengganu, Malaysia; 2Chemical Process Engineering Department, Malaysia-Japan International Institute of Technology, Universiti Teknologi Malaysia, Jalan Sultan Yahya Ahmad Petra, Kuala Lumpur 54100, Johor, Malaysia; kamyar@utm.my (K.S.); mahmoudeithar@cheme.utm.my (M.M.N.); 3Radiation Processing Technology Division, Malaysia Nuclear Agency, Kajang 43000, Selangor, Malaysia; m_yusof@nuclearmalaysia.gov.my; 4Department of Renewable Energy Engineering, Faculty of Chemical & Energy Engineering, Universiti Teknologi Malaysia, Johor Bahru 81310, Johor, Malaysia; rafiziana@utm.my

**Keywords:** silver nanoparticles, pullulan, gamma irradiation, green synthesis, antibacterial

## Abstract

The production of pure silver nanoparticles (Ag-NPs) with unique properties remains a challenge even today. In the present study, the synthesis of silver nanoparticles (Ag-NPs) from natural pullulan (PL) was carried out using a radiation-induced method. It is known that pullulan is regarded as a microbial polysaccharide, which renders it suitable to act as a reducing and stabilizing agent during the production of Ag-NPs. Pullulan-assisted synthesis under gamma irradiation was successfully developed to obtain Ag-NPs, which was characterized by UV-Vis, XRD, TEM, and Zeta potential analysis. Pullulan was used as a stabilizer and template for the growth of silver nanoparticles, while gamma radiation was modified to be selective to reduce silver ions. The formation of Ag-NPs was confirmed using UV–Vis spectra by showing a surface plasmon resonance (SPR) band in the region of 410–420 nm. As observed by TEM images, it can be said that by increasing the radiation dose, the particle size decreases, resulting in a mean diameter of Ag-NPs ranging from 40.97 to 3.98 nm. The XRD analysis confirmed that silver metal structures with a face-centered cubic (FCC) crystal were present, while TEM images showed a spherical shape with smooth edges. XRD also demonstrated that increasing the dose of gamma radiation increases the crystallinity at a high purity of Ag-NPs. As examined by zeta potential, the synthesized Ag-NP/PL was negatively charged with high stability. Ag-NP/PL was then analysed for antimicrobial activity against *Staphylococcus aureus*, and it was found that it had high antibacterial activity. It is found that the adoption of radiation doses results in a stable and green reduction process for silver nanoparticles.

## 1. Introduction

A multiphase material composed of at least one dimension below 100 nm is regarded as a nanomaterial. Polymer/silver nanomaterials that combine the advantages of the metal particles and the polymer’s processability open a new gateway in developing new nanocomposite systems with improved performances [1]. Investigation of metallic nanoparticles is a continuing concern within other nanomaterials. They are considered to be very promising as they contain remarkable antibacterial properties due to their large surface area-to-volume ratio, which is of interest for researchers due to the growing microbial resistance against metal ions and antibiotics, and the development of resistant strains [2,3]. Metallic nanoparticles, particularly silver, have a better property than that because of their bulk structure. Thanks to their flexible structure, their unique properties contribute to better thermal, electronic, antimicrobial, and sensing functionalities [4]. A previous researcher also reported that incorporating an organic material into inorganic particles will prevent the agglomeration of colloidal dispersions [5]. The aggregations of particles in colloidal dispersions are expected to reduce their diffusivity, thus limiting the contact with the bacteria.

Over the past century, there has been a dramatic increase in the exploration of the potential of metal nanoparticles (NPs). NPs are reported as promising materials due to their significant composition of high-energy surface atoms [6]. Factors that influence the higher content of high-energy surface atoms in NPs are believed to be based on their exceptional physical and chemical properties, which correspond to their bulk solid counterparts [7]. More recent attention has focused on the provision of producing silver NPs (Ag-NPs) because they have several superior properties and are widely used in different fields. Ag-NPs are used in diverse fields, including the medical, catalysis, electronic, optic, environmental, biotechnology, and packaging industries [8]. The antibacterial properties are the most encouraging characteristic in Ag-NPs as they provide an antimicrobial nature in the final product embedded with Ag-NPs.

Green synthesis of Ag-NPs always focuses on using starch, fungus, yeasts, plant extracts, and other biological and natural materials as a reactant [9,10,11]. The use of these materials in producing Ag-NPs has been stated as a safe and eco-friendly process due to the absence of hazardous reducing agents [12]. Pullulan, in this case, which is also regarded as a biomaterial, is derived from the polymorphic fungus *Aureobasidium pullulans* and has the structure of a linear homopolysaccharide of glucose [13,14]. Pullulan is composed primarily of maltotriose units, which are units of three α-1,4-linked glucose molecules that are polymerized linearly via α-1,6-linkages. Figure 1 illustrates the proposed structure of pullulan. The fabrication of Ag-NPs using pullulan is a better approach because the pullulan will act as a capping agent. According to previous researchers, these green-synthesized Ag-NPs, in comparison to chemically synthesized synthetic Ag-NPs, were found to be less toxic [15]. Several studies have reported that the toxicity of silver nanoparticles on normal cells varies depending on the particle size, capping agent and reducing agent used, and the techniques in synthesizing the nanoparticles [16]. It is known that toxicity of Ag-NPs towards normal cells is likely but minimal if synthesizing using a biomaterial. The reactivity of Ag-NPs to generate ROS was shown to induce the toxicity of Ag-NPs [15].

Pullulan is chosen as a biopolymer material to improve the reduction process without accelerating, reducing, or complexing agents. In addition, the structure of the polysaccharides in pullulan is reported to enhance the antibacterial activity by encapsulating the Ag-NPs, thus producing more stable nanoparticles [17]. This, in turn, will result in uniform and monodisperse nanoparticles being made [18]. Moreover, the particle size can also be tailored to the desired size with the incorporation of capping agents such as polymers, plant extracts, and, in this case, pullulan. It is reported that pullulan has a number average molecular weight (M_n_) of 90,000–150,000 daltons (Da) and a weight average molecular weight (M_w_) of 380,000–480,000 Da. The polydispersity index (PI) of pullulan ranges from 4.22 to 3.2 [13]. The capping agent molecules are believed to collide with the metal particles, which later induce the particles’ aggregation, shaping the particle size into the desired shape [19].

The fabrication of silver nanoparticles with the right shape and uniform size distribution inside the matrix, on the other hand, remains challenging. A radiation-induced process is ideal for generating metal particles in a solution, notably those of silver. Metallic ions are reduced at each interaction. Previous studies have shown that radiolytically produced species, solvated electrons, and secondary radicals have substantial reducing potentials [19]. In this study, pullulan was selected for generating silver nanoparticles (Ag-NPs) as it is non-toxic, renewable, and biocompatible [20,21,22,23,24,25,26]. The gamma irradiation synthesis method was introduced because it reduces the reaction time, produces a higher yield, and is green compared to the chemical reduction method [6]. The radiation method offers many advantages for the preparation of metal nanoparticles. Hydrated electrons, resulting from the aqueous solution’s gamma radiolysis, can reduce metal ions to zero-valent metal particles, avoiding the use of additional reducing agents and the consequent side reactions.

Traditional chemical methods for Ag-NP synthesis involve toxic chemicals (such as potent reducing agents, hydrazine, ethylenediaminetetraacetic acid, and, above all, sodium borohydride) and exposure to high-temperature conditions (such as in the citrate-based method). Therefore, they are not compatible with the green chemistry principle [27,28], which is why it is necessary to develop facile and green technologies in nanomaterial synthesis [29]. Therefore, this work addresses the need to synthesize silver nanoparticles in a green and clean process without the need for a chemical reducing agent. The gamma irradiation technique was chosen due to its sterile and inert process. In addition, this method also provides faster reaction times, higher yields, and improved material properties [14]. Hence, based on a previous study, Ag-NPs on pullulan were synthesized by the gamma irradiation method.

## 2. Materials and Methods

### 2.1. Materials

All materials and reagents used in this work were of analytical grade and used as received without further purification. Silver nitrate (AgNO_3_-99.85%) was used as the silver precursor and was obtained from Acros Organic (Carlsbad, CA, USA). Pullulan powder (R&M Chemicals, London, UK) was applied as a solid support for Ag-NPs. All these aqueous solutions were used with double-distilled water.

### 2.2. Methods

#### 2.2.1. Synthesis of Ag-NPs on Pullulan by Using γ-Irradiation

For the synthesis of Ag-NP/PL nanocomposites, 3.0 g of pullulan was dispersed in 100 mL double-distilled water and consistently stirred for 1 h at 90 °C until a clear solution was obtained. The solution of PL was left to cool at the ambient temperature; thereafter, 100 mL of the aqueous solution of AgNO_3_ (0.1 mol/L) was added, and the mixture was further stirred for 1 h. The mixture was then divided into six equal part (20 mL) sample bottles, purged by N_2_ for 30 min, and sealed. Finally, the suspension, which contained AgNO_3_/PL (A0), was irradiated under γ-irradiation with absorbed doses of 5, 10, 15, 20, 25, and 50 kGy (A1–A6). The γ-irradiation process was carried out in a ^60^Co Gammacell irradiator at room temperature (the dose rate was 35.7 kGy/min) provided by the Malaysia Nuclear Agency, Bangi.

#### 2.2.2. Characterization Methods and Instruments

The formation of Ag-NPs was confirmed using UV–Vis spectroscopy analysis. At a medium scan rate, the produced Ag-NPs were scanned from 300 to 1000 nm with a UV–Vis spectrophotometer (UV-2600 Shimadzu, Kyoto, Japan). The spectra of the sample were measured using a designed holder for a sample of 2 cm × 2 cm in dimension. It is known that UV–Vis analysis can be used to determine the peak of the metal atoms. The peak detection is due to the colored metal ions in response to the absorption of visible light. In addition, the electrons within the metal atoms exit one electronic state to another, leading to a strong peak presence at a specific range.

The crystallinity of the nanomaterials was determined by XRD analysis. In this analysis, the XRD pattern was scanned from 10° to 80° at a 2θ angle. The identity of the silver nanoparticles can be confirmed by comparing the XRD pattern with the library. The XRD samples were dropped until they reached a certain thickness in the thin film by being repeatedly dropped and dried at 60 °C. The structure of the produced Ag-NPs was examined using XRD-Empyrean (PANanalytical, Malvern, UK).

The TEM sample was prepared by dropping the Ag-NPs on the surface of a carbon-coated copper grid. This technique determined the morphology and size of the nanoparticles. The dried sample was scanned using a JEM-2100F transmission electron microscope (JEOL, Akishima, Japan).

The stability of A-NP/PL was determined by measuring the zeta potential using a particle analyzer (Nano-Plus Zeta Sizer, Tokyo, Japan). The instrument was attached to a He-Ne laser lamp (0.4 mW) at a wavelength of 633 nm. Measurements were performed at 25 °C in an insulated chamber with 10 runs for each measurement. Prior to the measurement, the colloidal suspension was sonicated for 10 min.

The microbial activity assay of the biofilm embedded with Ag-NP/PL against Gram-positive *Staphylococcus aureus* was carried out using the culture medium toxicity method. The biofilm sample with a dimension of 1 cm × 1 cm was placed on the surface of an agar plate and seeded with 100 μL of test culture consisting of microorganisms. Petri dish covers were sealed by wax to avoid any type of contamination. The Petri dish was incubated for 24 h, and the microbial growth was followed by visual observation. Natrium Agar (NA) was used as a medium with a temperature of 37 °C. The clear zone that formed around the film sample in the medium was recorded as an indication of inhibition against the microbial species.

## 3. Results and Discussion

### 3.1. Synthesis and Characterization of Silver Nanoparticles from Pullulan

The formation of Ag-NPs by reducing silver ions to silver metal species was achieved in the presence of aqueous solutions of pullulan or oxidized pullulan at room temperature over a period of 2 h with different gamma radiation doses. This process was monitored closely using UV–Vis spectroscopy. From the prepared samples, it can be seen that a different color intensity was observed depending on the gamma ray dosage, as shown in Figure 2. Ag-NP/PL suspensions without γ-rays (A0) showed a light brown solution and exhibited no Ag^+^ ion formation. As the γ-rays increased (A1–A6 with 5–50 kGy, respectively), the suspension marked a dark brown solution. It should be noted that the pullulan solution is transparent without color. The solution of Ag-NP/PL changed from a pale yellow to a dark brown solution after the gamma irradiation process was adopted.

### 3.2. Formation Mechanism of Silver Nanoparticles

In general, the radiolysis process involves producing a large number of homogenously distributed hydrated radicals [21,30,31,32,33]. Hydrated electrons and primary radicals and molecules appeared when the AgNO_3_/PL aqueous suspensions were exposed to γ-rays, as shown in Equation (1). Ag^+^ and ${\mathrm{NO}}_{3}^{-}$ were created during the separation reaction of the AgNO_3_ aqueous solution, as described in Equation (2). It is known that the $e_{aq}^{-}$ and H atoms are vital reducing agents; thus, they can be effectively reduced to the zero-valent state (Equations (3) and (4)) [34]. Larger clusters of silver atoms (Equation (6)) were produced due to the coalescence of silver atoms created during the irradiation process. Then, a stabilized Ag cluster was formed after the aqueous electrons reacted with the Ag+ groups (Equation (7)) [35,36]. Finally, a large number of aqueous electrons ($e_{aq}^{-}$) were provided after γ-irradiation, resulting in silver ions being reduced to Ag-NPs (Equation (8)). In addition, the incorporation of pullulan also created more hydrated electrons (OH-). The hydrated electrons can reduce all silver ions to silver atoms, ending up as a nucleus for the successively formed atoms. The proposed mechanism of the formation of the silver nanoparticles is illustrated in the graphic in Figure 3.

The pullulan matrix is expected to better interact with the nanoparticles by preventing particle agglomeration between molecules [1,37]. As shown in Figure 3, the silver nanoparticles were coated with pullulan, indicative of interaction between multiple hydroxyl groups brought by pullulan. These, in turn, will arrange the silver by encapsulating the nanoparticles to form stable colloidal suspensions.
(1)$$H_{2}O\overset{\gamma -Ray}{\to }e_{aq}^{-}+H^{⦁}+OH^{⦁}+H_{3}O^{+}+H_{2}+H_{2}O_{2}+\ldots$$
(2)$$AgNO_{3}\overset{yields}{\to }\;Ag^{+}+NO_{3}^{-}\;$$
(3)$$Ag^{+}+e_{aq}^{-}\;\overset{reduction\;}{\to }\;Ag^{o}$$
(4)$$Ag^{+}+H^{⦁}\;\overset{reduction\;}{\to }\;Ag^{0}+H^{+}\;$$
(5)$$Ag^{0}+Ag^{+}\;\overset{}{\to }\;Ag_{2}^{+}$$
(6)$$nAg^{+}+Ag_{2}^{+}\;\overset{}{\to }\;{\left(Ag\right)}_{n}^{+}$$
(7)$${\left(Ag\right)}_{n}^{+}+ne_{aq}\overset{}{\to }\;{\left(Ag\right)}_{n}$$
(8)$${\left(Ag\right)}_{n}\;\overset{\gamma -Ray}{\to }\;{\left(Ag\right)}_{n}^{+}+ne_{aq}^{-}$$

### 3.3. UV–Vis Analysis

UV–Vis analysis is a top screening analysis in confirming the formation and strength of silver nanoparticles. In this study, the appearance of Ag-NPs with different gamma radiation doses was monitored using UV–Vis spectroscopy. Figure 4 shows the UV–Vis absorption spectra of Ag-NP/PL at different gamma ray doses. The figures show that the highest-intensity absorption spectra exhibited a similar pattern for all radiation doses, known as surface plasmon resonance (SPR), at 410–420 nm. This result reveals that Ag-NP/PL prepared at 50 kGy, 25 kGy, 20 kGy, 15 kGy, 10 kGy, and 5 kGy produced absorption spectra with sharp peaks (surface plasmon resonance, SPR) at 410–420 nm. On the other hand, in Ag-NP/PL prepared without gamma radiation, the SPR peaks did not appear at 420 nm, indicating that no Ag-NPs were produced.

At a higher intensity, the gamma radiation provided a more spherical shape with a smaller size of produced NPs. For instance, at 50 kGy, the absorbance peak was at 0.9 compared that at 25 kGy, which was at 0.4. It can be depicted that the highest absorbance reflects a highly uniform particle size [38]. For reference, Ag-NP/PL was also prepared without the gamma radiation method. There was no absorption peak recorded at 420 nm, implying that Ag-NPs were not successfully produced. Despite this, the UV–Vis analysis also revealed that the shape and size of nanoparticles, which are dependent on the wavelength of absorption in the UV spectrum, were indicatively affected by the plasmon resonance. As shown in Figure 4, the spectra of silver nanoparticles are consistent with the spherical shape of different size Ag-NPs, which can later be proved in TEM analysis [39].

### 3.4. XRD Analysis

In order to further investigate the reaction mechanism that occurred during the synthesis, XRD analysis was carried out to determine the purity and crystallinity of Ag-NP/PL, as presented in Figure 4. It is known that pullulan (PL) exists in an amorphous state [40]. The incorporation of Ag-NPs and irradiation techniques was previously shown to transform the solution to exhibit crystallization characteristics [41]. Four (4) diffraction peaks can confirm the purity of the silver formation at 2θ of 38.54°, 44.91°, 65.04°, and 78.11°, corresponding to the (111), (200), (220), and (311) planes of the face-centered cubic (FCC) structure [42]. These four notable diffraction peaks are in good agreement with the theoretical standard figures (JCPDS file no. 01-087-0718) [42]. The formulation samples of A1–A6 in Figure 4 indicate that only silver metal is present in their crystalline phase without any significant traces of other substances. In addition, the occurrence of a nanosized particle of Ag-NP/PL promotes the considerable traces of silver formation in the XRD pattern peaks, as shown in A6 of Figure 5.

### 3.5. TEM Analysis

Transmission electron microscopy (TEM) imaging was conducted to evaluate the morphology and particle size distribution of Ag-NPs. The TEM micrograph clearly reveals that the silver nanoparticles were well dispersed in the pullulan matrix, as shown in Figure 6, Figure 7, Figure 8 and Figure 9, and this result is in line with that of previous studies [14]. In addition, the Ag-NPs were also segregated evenly with almost no agglomeration. Spherical and oval shapes were also observed in the TEM images [43]. Studies by Yoksan and Chirachancai (2010) showed that the particle size distribution (mean diameter and standard deviation) was drastically decreased to 3.98 ± 1.356 nm at 50 kGy doses, indicating an even Ag-NP segregation process [44]. As mentioned previously, in UV–Vis analysis, it can be stated that the γ-rays reduce the particle size distribution during the irradiation process. This condition confirms that the formation of the particle size distribution (mean diameter of particles) depends on the γ-ray doses. For low and medium γ-ray doses, i.e., 5, 10, and 25 kGy, the mean diameter and standard deviation of Ag-NPs were noted as 40.97 ± 31.64, 18.52 ± 13.906, 9.84 ± 3.595 nm, and 3.98 ± 1.356, respectively.

The combination of nanostructures at different sizes leads to the broadness of the size distribution peaks, as shown in Figure 6, Figure 7, Figure 8 and Figure 9, which proves that the particles at high γ-doses were highly uniform and homogenous. In addition, 50 kGy γ-ray doses offer a more acceptable size (finer) of silver nanoparticles, indicating that hydrated radicals (ē aq) are entirely produced; thus, this promotes the reduction of silver ions to silver atoms [30]. Moreover, the Ag-NPs had a narrow size distribution, implying that they were very uniform at 50 kGy (Figure 9). These results also reveal that as the gamma irradiation dosages were increased, the intermediate structure of the pullulan suspension significantly enhanced the interactions of $(e_{aq}^{-}$) electrons within the dissolved molecules, leading to an increased yield of nanoparticles. It is also found that pullulan also functions as a stabilizer during the synthesis of silver nanoparticles and as a capping agent by providing the template in the radiolysis process.

### 3.6. Antimicrobial Activity of Ag-NP/PL

It is vital to understand the stability of nanoparticles in aqueous solutions because the zeta potential is an indicator of the surface charge potential, which is a critical characteristic. The zeta potentials of Ag-NP/PL were found to be −72.05 ± 0.93 mV, as shown in Figure 10. It has been claimed that the generated nanoparticles have a negative charge on their surfaces. Having a positive or negative charge on the surface of nanoparticles makes them more stable and prevents them from aggregating together by pushing the same charges in the same direction [45]. Previous studies have confirmed that Ag-NPs can be considered stable if the zeta potential values are more than +30 mV or lower than −30 mV [45,46]. The zeta potential values recorded imply that the formation of Ag-NPs at the level of molecule activity led to stable forms of the nanoparticle solutions.

### 3.7. Antimicrobial Activity of Ag-NP/PL

The antimicrobial activity assay of the Ag-NP/pullulan biofilms against Gram-positive *Staphylococcus aureus* (*S.aureus*) was carried out by the culture medium toxicity method [47]. Typical broth and agar were used as media to grow the bacteria. The activity was evaluated after 24 h of incubation at 37 °C. The contact areas of all irradiation sample (0, 10, 25, and 50 kGy) biofilms were transparent, indicating inhibition of bacterial growth. The clear zone was prominent, as shown in Figure 11. At higher irradiation doses (50 kGy), the clear zone diameter was higher (16 mm) compared to that of the 25 kGy (12 mm) γ-doses.

The transparent zone was translated to the average diameter zone, which is tabulated in Table 1. Although the clear site was seen surrounding the biofilm embedded with Ag-NP/PL for both γ-ray doses, the microbe resistance was less effective for lower gamma rays. In short, the biofilm embedded with silver nanoparticles exhibited inhibitory activity against *S.aureus.* This antibacterial characteristic is expected to be created by the silver nanoparticles, which rupture the bacteria cell wall membrane. Disruption of the bacteria cell membrane instantly disturbs the respiration of the microbes, thus stopping the activity of the bacteria [48]. The outcomes of this analysis indicate that Ag-NP/PL embedded in the biofilm has good potential for antimicrobial food packaging applications.

The antibacterial activity of Ag-NP/PL was primarily affected by the silver ions (Ag^+^) that Ag-NPs released. As shown in Table 1, the higher the γ-ray doses, the better the antibacterial activity of the produced Ag-NPs. The release of more Ag^+^ is influenced by the size of the particles [7]. In short, the smaller the particle size of Ag-NPs, the more they release Ag^+^ due to the high surface area of Ag-NPs [49]. The antibacterial properties of Ag-NP/PL act by attachment of Ag-NP/PL on the cell membrane wall, leading to its rupture. Thus, Ag^+^ is discharged inside the cell, which retards the cell’s respiratory system by inducing the production of reactive oxygen species (ROS). An illustration of the mechanism of antibacterial activity that takes place with the aid of Ag^+^ is shown in Figure 12. The stress placed on the cell wall membrane affected by the increase in ROS production is an effective mechanism of Ag-NP-induced silver ions. A previous study also revealed that the permeation of silver ions into bacterial cells changes DNA molecules into a condensed form and prevents the ability of the cells to replicate [50]. Moreover, bacterial sterilization is also caused by the silver ions due to the reaction with proteins, which leads to the direct binding with the sulfhydryl group (-SH) and causes the loss of activity of multiple enzymes [51].

## 4. Conclusions

Pullulan-capped silver nanoparticles were successfully synthesized using the gamma irradiation technique. Stable Ag-NP/PL nanocomposites with an average size of 3.98 nm were prepared without any reducing agent. UV–visible spectroscopy confirmed the formation of silver nanoparticles by detecting a plasmonic band at 410–420 nm. The XRD pattern showed that the crystalline structure of the Ag-NPs for all samples was fcc. TEM imaging ascertained that the Ag-NPs were well dispersed in the pullulan matrix, with the particle diameter of the Ag-NPs gradually decreasing after higher doses at 50 kGy due to the γ-induced fragmentation in Ag-NPs. The zeta potential of Ag-NP/PL was negatively charged, and it was found to be a stable and good dispersion in a colloidal suspension. The biofilm embedded with Ag-NP/PL was found to have antibacterial activity against *Staphylococcus aureus*. In summary, pullulan-capped silver nanoparticles can be applied in a wide range of applications, particularly in antimicrobial biofilm packaging.

## Figures and Tables

**Figure 1 polymers-13-03578-f001:**
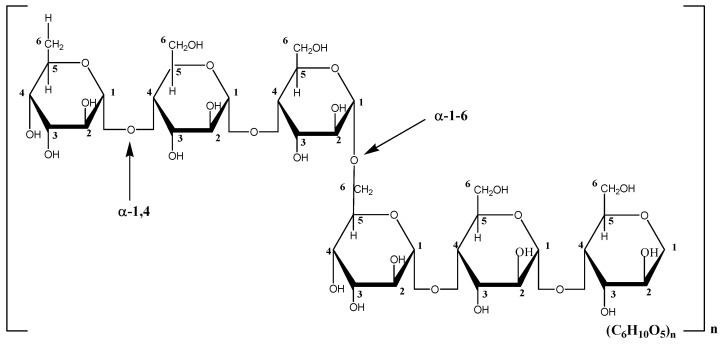
Structure of pullulan.

**Figure 2 polymers-13-03578-f002:**
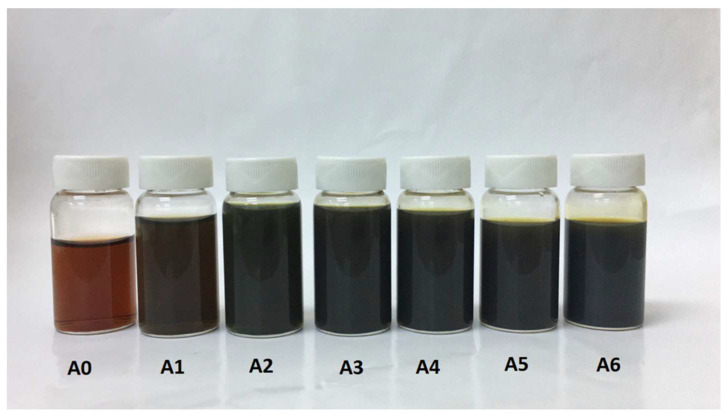
Photograph of Ag-NP/PL at different different γ-irradiation doses: 0, 5, 10, 15, 20, 25, and 50 kGy (**A0**–**A6**).

**Figure 3 polymers-13-03578-f003:**
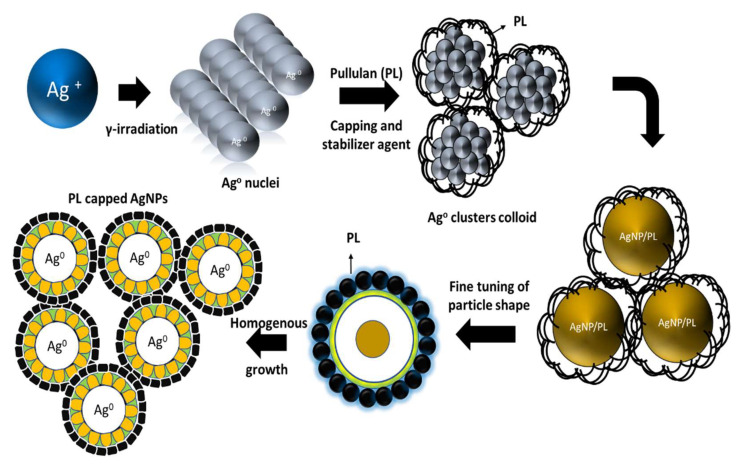
Proposed schematic mechanism of the synthesis and growth of silver nanoparticles on pullulan.

**Figure 4 polymers-13-03578-f004:**
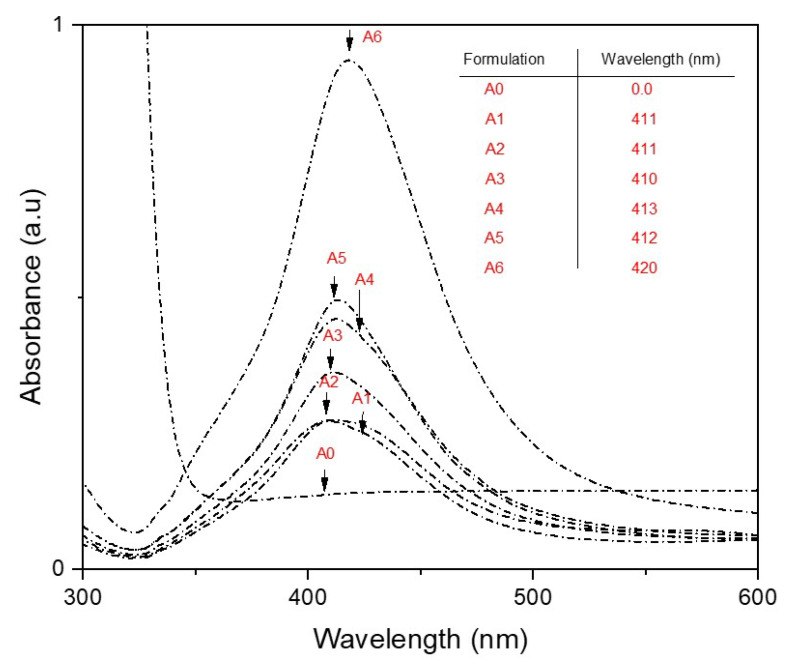
UV–Vis spectrum of Ag-NP/PL prepared at 0, 5, 10, 15, 20, 25, and 50 kGy gamma doses.

**Figure 5 polymers-13-03578-f005:**
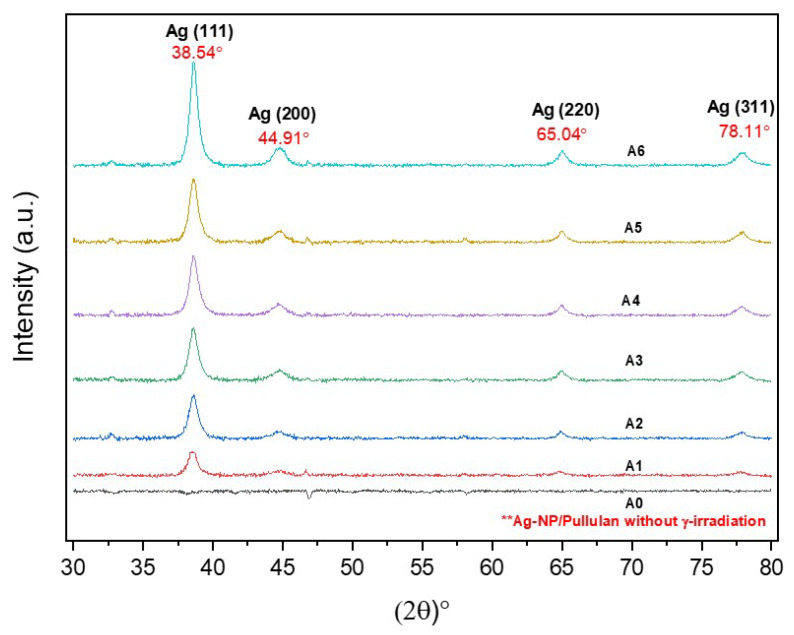
XRD patterns of Ag-NP/PL at different γ-irradiation doses: 0, 5, 10, 15, 20, 25, and 50 kGy (**A0**–**A6**).

**Figure 6 polymers-13-03578-f006:**
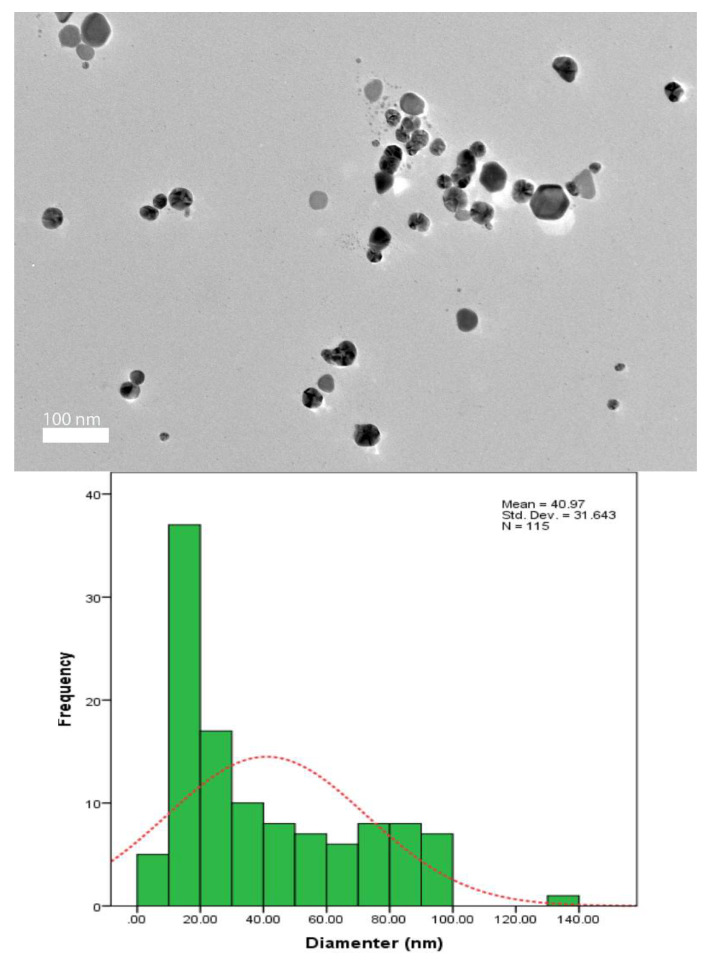
TEM image and corresponding particle size distribution of Ag-NP/PL at 5 kGy.

**Figure 7 polymers-13-03578-f007:**
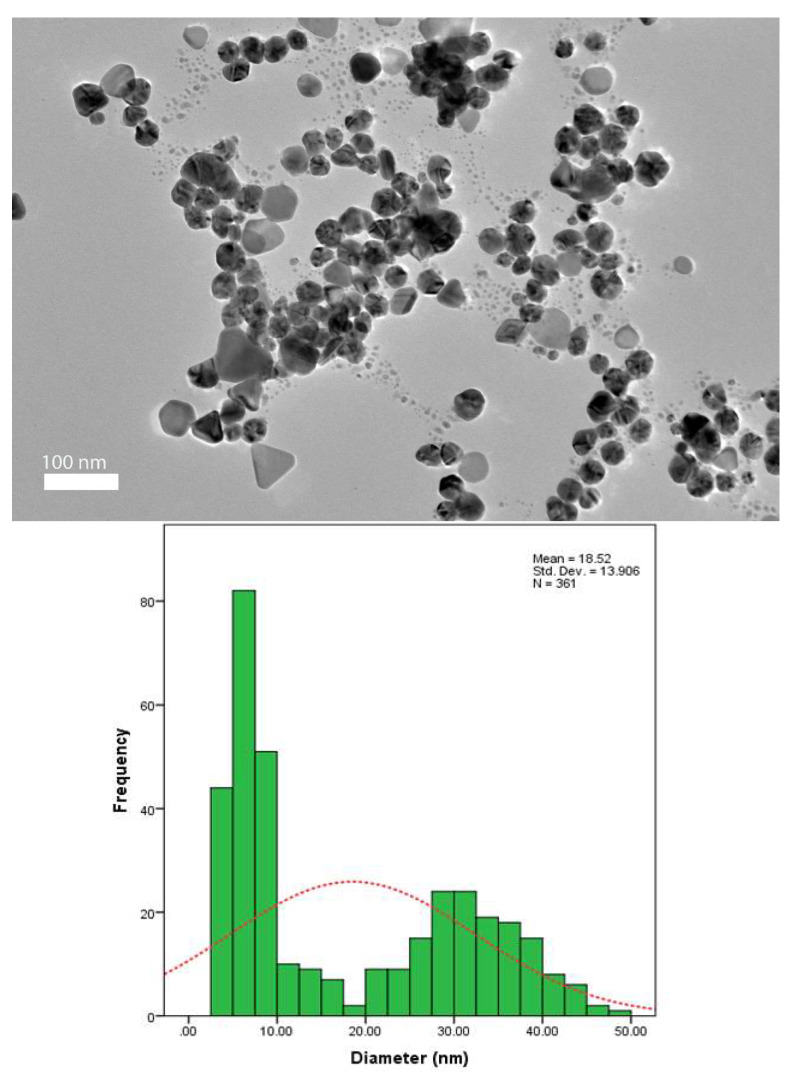
TEM image and corresponding particle size distribution of Ag-NP/PL at 10 kGy.

**Figure 8 polymers-13-03578-f008:**
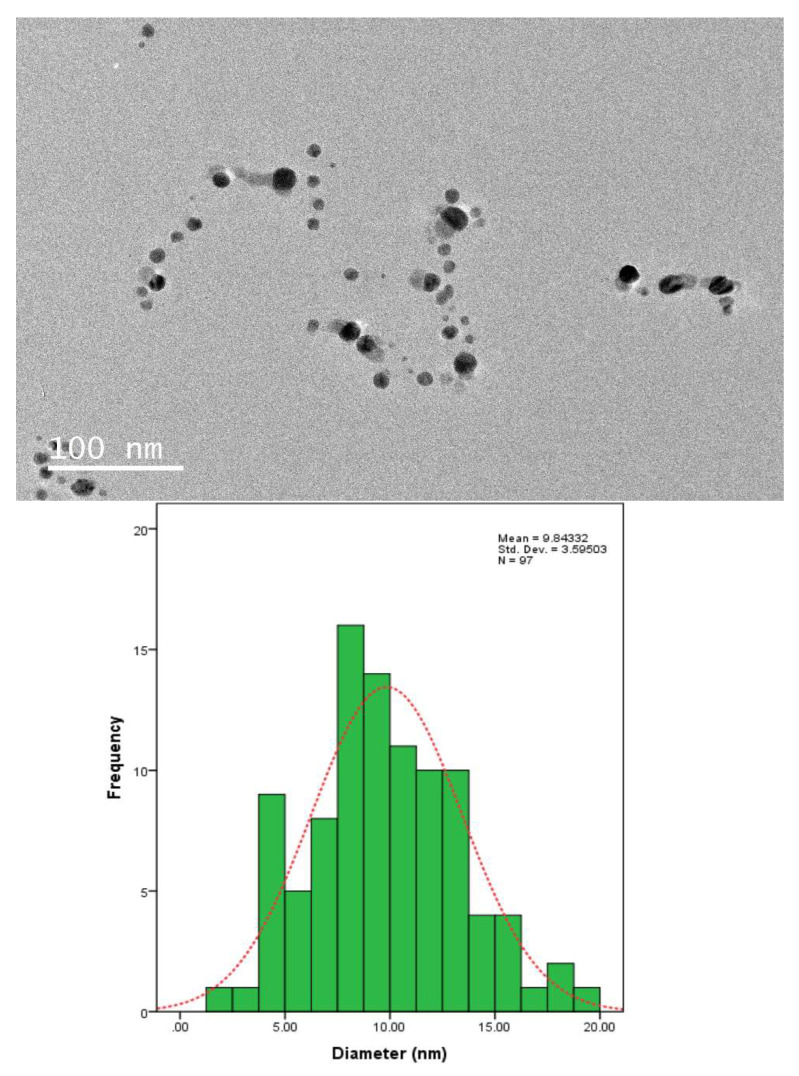
TEM image and corresponding particle size distribution of Ag-NP/PL at 25 kGy.

**Figure 9 polymers-13-03578-f009:**
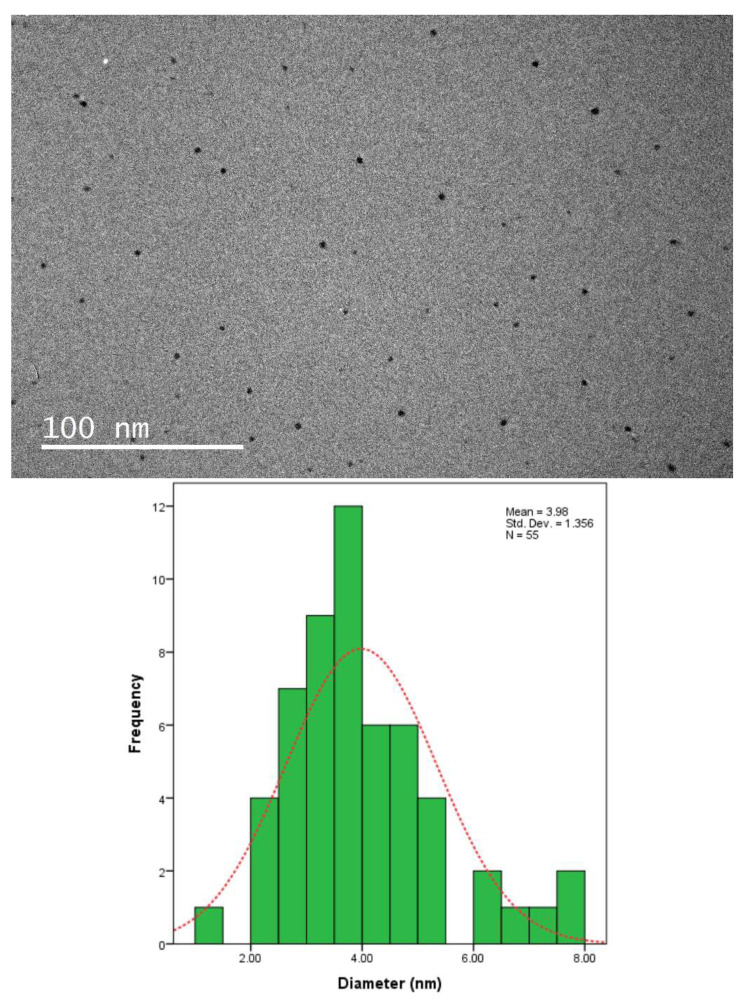
TEM image and corresponding particle size distribution of Ag-NP/PL at 50 kGy.

**Figure 10 polymers-13-03578-f010:**
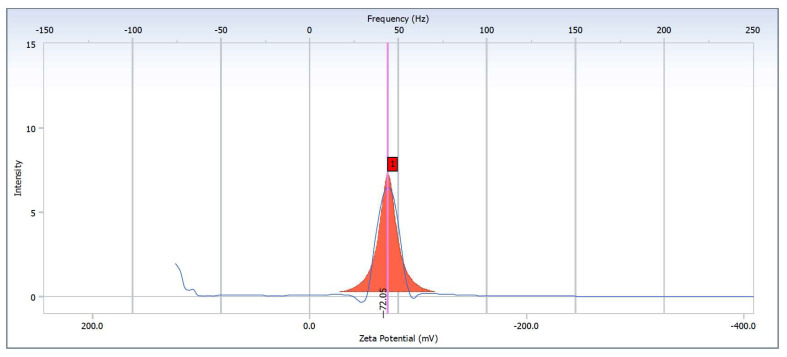
Zeta potential measurements for Ag-NP/PL at 50 kGy.

**Figure 11 polymers-13-03578-f011:**
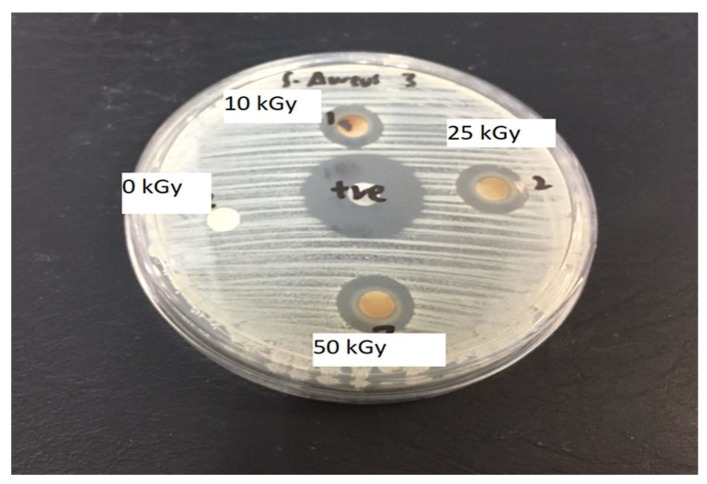
Photograph of antimicrobial test results against *Staphylococcus aureus*—After exposure to microbes.

**Figure 12 polymers-13-03578-f012:**
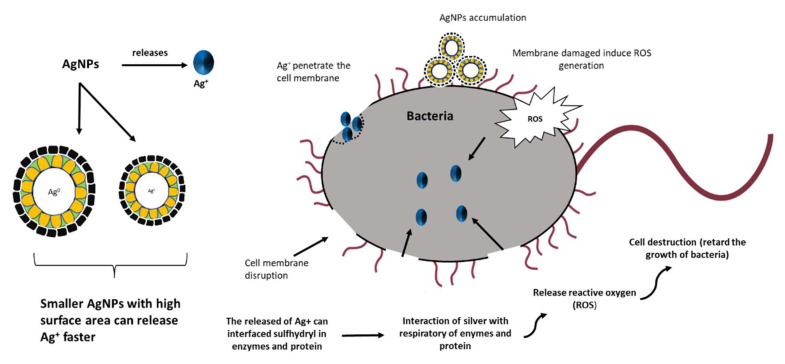
Antibacterial mechanism of Ag-NPs.

**Table 1 polymers-13-03578-t001:** Average diameter of inhibition zone.

Sample	Average Diameter of Inhibition Zone (mm)
0 kGy	0 ± 0
10 kGy	10 ± 1.65
25 kGy	11 ± 2.34
50 kGy	13 ± 2.01

## Data Availability

Not applicable.

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
