# Peer review of "Interaction Insight of Pullulan-Mediated Gamma-Irradiated Silver Nanoparticle Synthesis and Its Antibacterial Activity"

_polymers, 2021, doi:10.3390/polym13203578_

Round 1

Reviewer 1 Report

Dear colleagues,

The article entitled “Interaction Insight of Pullulan Mediated on Gamma-irradiated Silver Nanoparticles Synthesis and its Antibacterial Activity” explores the synthesis of AgNPs by resorting to gamma irradiation as reducing agent and the natural polymer pullulan as capping agent, thus obtaining homogeneously distributed ultrasmall AgNPs, with interesting antibacterial properties. The document is very interesting and well-written. Content is new, but reference to your conference paper is essential (doi: 10.1088/1757-899X/808/1/012030). You should compare your manuscript to the work presented in the conference paper, highlighting the differences, and novelty.

Some detailed suggestions can be found below:

Abstract:

starch-like substance? It would be better to indicate the origin of pullulan or highlight its main attractive feature for your study.

region of 420435 nm -> region of 420 - 435 nm

As observed by TEM images, it can be said that with increasing radiation dose, the particle size decreases, resulting in a mean diameter of the Ag-NPs ranging from 40.97 to 3.98 nm.

“In addition, Ag-NP/ PL has the potential to have high antibacterial activity.” -> I would delete this sentence.

It has been found that the adoption of radiation doses results in a stable and safe reduction process for silver nanoparticles. – if you write “safer” you should perform comparative studies to other synthesis methodologies.

Introduction

Lines 36-46: the choice of adding organic material to your inorganic particles could be better justified. One simple reason is that it has been reported that metal nanoparticles tend to form agglomerates when in colloidal dispersions, which reduces their diffusivity, limiting the contact with the bacteria.

Lines 64-69: pullulan will act as a capping agent. And reducing agent? Why pullulan?

Figure A2: the Figure is really nice, congratulations. I simply suggest improving it a little further, as the last two steps are not so clear. Suggestion: use a color for each element and identify each one, with the letters having the same color.

The particles done at 50 kGy γ-ray doses are gorgeous. Really nice work.

Do you have images of Ag-NP, having no pullulan included in the processing? You should also show the control built at 0 kGy γ-ray doses. That is important to reinforce your discussion from lines 246-251, and justify your title. Your focus is on the gamma irradiation-mediated process to obtain improved AgNPs. The coating with pullulan is justified theoretically in the beginning of the manuscript, but then you don’t do any comparative studies (with/without pullulan).

Line 263: S. aureus, not s. aureus.

Conclusion :

“The stable Ag-NP/PL with an average size of 10 nm were prepared without any reducing agent.” – why do you solely highlight the particles done at 25 kGy γ-ray doses? Which are the advantages and disadvantages of that one, compared to the other ones, namely the ones produced using higher doses?

Reviewer 2 Report

In this contribution by Salleh and co-workers, the authors prepared pullulan-mediated silver nanoparticles for antibacterial application. The results are kind of interesting and potentially attractive to the readership of Polymers. However, It could be publishable in due course but these points below must be addressed prior to publication.

  1. Line 23, ‘420435 nm’ should be ‘420-435 nm’.
  2. Line 58-59, several recent studies (doi.org/10.1016/j.actbio.2020.02.044; doi.org/10.1016/j.colsurfb.2020.111112; doi.org/10.1021/acs.biomac.9b01724) should be included to support such claim.
  3. What’s the molecular weight and polydispersity of pullulan? As a polymer journal, such information is a must.
  4. The resolution of figure 5-8 should be improved to a higher level.
  5. What’s the stability of these silver nanoparticles in physiological medium?
  6. Does this silver nanoparticle have toxicity on normal cells?

Round 2

Reviewer 1 Report

Dear Authors,

Very nice work. Thank you for your care in revising your manuscript. 

In my opinion, the manuscript is now fit to be published in Polymers.

Best of luck in your future work.

Reviewer 2 Report

I recommend it for publication. At the same time, the authors should revise the citation style during the proof stage according to the request of polymers.

  1. Most of the reference is with '[Internet]'.
  2. The volume and page information for Ref. 16, 44, 47, and 48 are missing. Please check all.